# The Role of DAMPS in Burns and Hemorrhagic Shock Immune Response: Pathophysiology and Clinical Issues. Review

**DOI:** 10.3390/ijms22137020

**Published:** 2021-06-29

**Authors:** Desirè Pantalone, Carlo Bergamini, Jacopo Martellucci, Giovanni Alemanno, Alessandro Bruscino, Gherardo Maltinti, Maximilian Sheiterle, Riccardo Viligiardi, Roberto Panconesi, Tommaso Guagni, Paolo Prosperi

**Affiliations:** 1ESA-European Space Agency Headquarter, 24 Rue de Général Bertrand, 75345 Paris, France; 2Department of Experimental and Clinical Medicine, University of Florence, 50121 Firenze, Italy; 3Trauma Team, Acute Care Surgery and Trauma Unit, Careggi University Hospital, Largo A. Brambilla 3, 50134 Florence, Italy; bergaminic@aou-careggi.toscana.it (C.B.); martelluccij@aou-careggi.toscana.it (J.M.); alemannog@aou-careggi.toscana.it (G.A.); bruscinoa@aou-careggi.toscana.it (A.B.); maltintig@aou-careggi.toscan.it (G.M.); max.scheiterle@gmail.com (M.S.); viligiardir@aou-careggi.toscana.it (R.V.); panconesir@aou-careggi.toscana.it (R.P.); tommaso.guagni@gmail.com (T.G.); prosperip@aou-careggi.toscana.it (P.P.)

**Keywords:** burns, DAMPs, alarmin, trauma, cytokine production, shock, hemorrhagic shock

## Abstract

Severe or major burns induce a pathophysiological, immune, and inflammatory response that can persist for a long time and affect morbidity and mortality. Severe burns are followed by a “hypermetabolic response”, an inflammatory process that can be extensive and become uncontrolled, leading to a generalized catabolic state and delayed healing. Catabolism causes the upregulation of inflammatory cells and innate immune markers in various organs, which may lead to multiorgan failure and death. Burns activate immune cells and cytokine production regulated by damage-associated molecular patterns (DAMPs). Trauma has similar injury-related immune responses, whereby DAMPs are massively released in musculoskeletal injuries and elicit widespread systemic inflammation. Hemorrhagic shock is the main cause of death in trauma. It is hypovolemic, and the consequence of volume loss and the speed of blood loss manifest immediately after injury. In burns, the shock becomes evident within the first 24 h and is hypovolemic-distributive due to the severely compromised regulation of tissue perfusion and oxygen delivery caused by capillary leakage, whereby fluids shift from the intravascular to the interstitial space. In this review, we compare the pathophysiological responses to burns and trauma including their associated clinical patterns.

## 1. Methods

The literary search for this topic was performed using the PRISMA checklist through the MEDLINE, EMBASE, and GOOGLE SCHOLAR databases using the following keywords: “DAMPs in burns”, “DAMPs in trauma hemorrhagic shock”, “hypovolemic shock in burns”, “hemorrhagic shock”, “diagnostic applications and DAMPs”, and “therapy and DAMPs”. Relevant references were searched and included review articles. 

Collected papers were selected according to the following criteria: publications had to be in English, or, if in another language, be provided from online translation tools or at least come in the form of a structured English abstract. Only studies conducted on an adult population were considered.

## 2. Introduction

Burns are a peculiar section of trauma. This review refers to the adult population. For both burns and hemorrhagic shock following a major trauma, the ages of higher incidence are similar, between the 20 to 40 years, mainly males (the data refer to high income countries [1,2,3]). Burns are the fourth leading cause of traumatic events in the world. After traffic accidents, falls, and violence. Hemorrhage following trauma accounts for 40% of deaths, as any patient with multiple traumas can have some degree of hypovolemia and can potentially be affected by ongoing severe bleeding (ATLS) [4], but ATLS can be also present in an injury involving only one area (for example abdominal trauma with severe liver injury [4]). Burns can cause a decrease in the quality of life, increased disability, and death [1,5]. The most prevalent age for burns is between 20 and 30 years. There is a wide variability in the incidence of burn injuries across the world. For example, the number of burn- related deaths per 100,000 population ranges from 0.02 to 14.53 according to the geographical area [1,5]. The World Health Organization estimates that annually over 265,000 deaths result from fire-related burns, with over 95% occurring in low and middle income countries [6,7]. Regarding Europe (European Health Statistics) [8], Brusselaers et al. [2] in a systematic review reported that the annual incidence of severe burns was 0.2 to 2.9/10.000 inhabitants with a decreasing trend in time. The causes of burns are numerous and include friction, thermal (cold and heat), chemical, and electrical [9]; the majority of burn injuries are caused by heat from hot liquids, solids or fire. In burns, consequences are commensurate to the extent of the inflammatory response [4,9,10] and the inflammation severity is strictly associated with burn severity, i.e., extension and depth [4,9,10]. However, the most widely used classification for predicting morbidity and mortality [9,10] is the percentage of the total body surface area involved. (TBSA). Burns can be minor or major depending on extension and depth. Minor burns are usually superficial, encompassing <20% of the total body surface area (TBSA), while major burns are deeper and usually classified as >10% TBSA in elderly patients, >20% TBSA in the adult population, and >30% TBSA in children [9,10]. Lesions of the respiratory apparatus due to the inhalation of various types of fumes and vapors must be also considered, as they may require preventive intubation. An airway compromise from massive edema after burn injury should be suspected even if it is not immediately evident, as it frequently does not manifest in the first 24 h [4,10]. Traumatic hemorrhagic shock is the most common cause of death and remains the leading cause of death in persons aged 1–44 years [4]. In a paper in 2016, Haagsma et al. [11] reported that trauma is the sixth leading cause of death in the world. According to the global burden of disease (GBD) [1,2,4,11], currently, 5.8 million people of all ages and economic groups die every year as a result of unintentional injuries and violence. Injuries account for 18% of the total disease burden worldwide. In the adult population, special attention must be paid to people of advanced age (>65), as the ageing process may affect many aspects of the normal physiology: receptor response to catecholamines, cardiac compliance, inability to increase the heart rate, or the efficiency of myocardial contraction when stressed by blood volume loss. Atherosclerotic vascular disease may also be present, making vital organs extremely sensitive to the reduction in blood flow. Other groups that need special attention are athletes from the changes in their cardiovascular dynamics and pregnant women in whom a physiological hypervolemia is present (ATLS) [4]. 

Damage-associated molecular patterns (DAMPS) play a role in the pathophysiology of the post-injury systemic inflammatory response that begins immediately after injury [3,12,13]. This response is present in trauma, surgical, and critically ill patients, although in burns it has a peculiar and unique severity and magnitude [14,15].

DAMPs are expelled after tissue damage, and are immunologically active and clinically relevant [3]. Some DAMPs are intracellular proteins from different organelles that are released after cellular death (for example high mobility group box 1, HMGB1); others are non-protein (ATP, DNA), extracellular matrix, or cell membrane components secreted after host recognition of tissue damage [3]. When released in the extracellular environment, they are able for activating innate and adaptive immunity. The elicited local inflammatory response has the goal of promoting adequate tissue repair, but it can also activate an uncontrolled systemic response capable of inducing remote organ failure (MOF) [9,12,13,14]. The DAMPs hyperinflammatory response is associated with a counterbalancing compensatory anti-inflammatory response syndrome (CARS), resulting in post-traumatic immunosuppression (IS) [9,12,13]. These processes are believed to occur simultaneously rather than sequentially, as was thought until a few years ago [12,13]. A slight delay between initial hyperinflammation and the following hyper-resolution is likely to be present, due to the production of suppressing/inhibiting inducible DAMPs (SAMPs) by DAMPs-activated/initialized responses of innate immune cells [9,12,13,15,16].

DAMPs responses are present in both burns and trauma although the hypermetabolic status that develops after major burns is a unique condition not present in trauma [9,10,14]. This review examines the role of DAMPs in two types of responses to injury: burns and traumatic hemorrhagic shock. Suggestions for the use of DAMPs as biomarkers for both conditions are reported, as well as their possible role in modulating immune responses and in limiting excessive inflammatory responses. 

A large number of DAMPs, or “alarmins” have been described to initiate and perpetuate the systemic post-traumatic and/or noninfectious inflammatory response [12]. The innate immune system has been long considered to be able to respond to the “infectious non self”, but it has now proven to be able to also respond to signs of destruction or immunogenic cell death through host intra- and extracellular material [17]. Candidate DAMPs must have the following characteristics: receptor agonism at physiological doses and release from cells under stress or cell death [17]. They are mitochondrial and nuclear nucleic acids, histone-associated proteins, adenosine triphosphate (ATP), heat shock proteins (HSPs), and others [18].

Any kind of traumatic event/tissue injury provoked by mechanical trauma, thermal trauma, or metabolic trauma, such as that promoted by ischemia/reperfusion injury, acidosis, and hypoxia/hypoxemia, provokes a DAMPs release [12,13]. This sterile response serves as a defense strategy aimed at maintaining and restoring homeostasis. When DAMPs are released in the extracellular environment, they are able to activate innate and adaptive immunity. Innate immunity cells such as antigen presenting cells (APC), dendritic cells (DCs), or neutrophils (PMNs) have pattern recognition receptors (PRRs) capable of recognizing DAMPs. After PRRs, PMNs, and APC activation, the production of cytokines, chemokines, and other soluble factors starts [19]. This local inflammatory response is meant to elicit adequate tissue repair, but it may also produce an uncontrolled systemic response that is able to induce remote organ failure. 

A general “inflammatory pathway” was described in 2008 [20,21]. In this pathway the DAMPs can be grouped under the following categories: *inducers, sensors, mediators, and effectors.*

*Inducers* initiate the inflammatory response. They include molecules produced by damaged cells (DAMPs) or by bacteria and viruses (pathogen-associated molecular patterns: PAMPs). The released DAMPs of the damaged cells, such as mitochondria and other molecules, for example, the high mobility box group 1 (HMGB 1), initiate the inflammatory response. In bacteria, the most well-known PAMPs are lipopolysaccharides (LPS) from gram-negative bacteria cell walls. For viruses, the most common PAMPS are viral DNA and RNA.

*Sensors* are the cell receptors (pathogen recognition receptors-PPRs) that recognize DAMPs or PAMPs. The most commonly known is the toll-like receptor-4 (TLR4) that binds LPS and initiates the inflammatory response. 

*Mediators* and *Effectors*: After this process of binding is completed, the signaling route starts leading to gene expressions to produce *mediators*, i.e., “cytokines”, tumor necrosis factor-α (TNF-α), interleukin-1 (IL-1), and interleukin-6 (IL-6). At the end of the process, *effectors* are the responding cells, tissues, and organs. For example, over a certain threshold, 15% of the total body surface area (TBSA) in burns, cytokines “spill” into the systemic circulation and act on the cells, tissues, etc. [20,21].

## 3. Indicators for Burns Shock and Traumatic Hemorrhagic Shock (Comparison)

Burn shock and traumatic hemorrhage show peculiar differences. In burns, edema is present and forms within the first hour after injury. This hypovolemic distributive shock is related to the leakage of fluid in the interstitial space and persists even after hypovolemia is corrected. A vicious cycle is engaged with cardiovascular dysfunction, that exacerbates the systemic inflammatory response (see Jeschke 2020 for review of literature [9,22]). The vascular permeability contributes to the formation of tissue edema in the burnt area and in distal organs and tissues. This capillary leakage due to the oxidative reaction increases the levels of nitric oxide and inflammatory mediators, damaging the vascular endothelium [9,23,24]. The fluid extravasation into injured and non-injured tissues takes place within the first 24–48 h from the harmful event. 

Hemorrhage is the most frequent cause of shock after trauma. In their review, Kushimoto et al. [25] reported that bleeding accounts for 30–40% of all trauma-related deaths that occur within hours after injury [26]. In hemorrhagic shock, the severe blood loss leads to inadequate oxygen delivery at the cellular level and if it continues unchecked, death quickly follows.

Burns lead to depressed cardiac function within a few hours, caused by oxidative stress and the release of IL-6. Tachycardia is also present, and it is related to the amount of blood loss.

Burns can also produce a hypermetabolic state (flow phase) [9,22] that is a peculiar and complex response that involves the release of stress hormones and pro-inflammatory mediators. This state may persist for 36 months after the initial injury. The systemic response linked to the hypermetabolic status that is produced is related to stress mediators, catecholamines, glucocorticoids, and glucagon. An impaired abnormal wound repair is associated with an increased number of mast cells (MC) that, after trauma, secrete histamine and degranulate instantaneously after thermal injury. An increase of reactive oxygen species (ROS) has also been shown to have a detrimental effect [27]. 

An increase in resting energy expenditure, temperature, protein loss, muscle wasting, and synthesis of acute phase proteins, for example insulin-like growth factor 1 (IGF-1), is present. These events lead to organ catabolism associated with organ disfunction that can lead to death [9]. 

Hepatomegaly is also present due to the presence of lipolytic products in adipose tissue that are toxic for the liver because it cannot metabolize them. The occurrence of hyperlipidemia and insulin-resistant hyperglycemia are present, which worsen the hypermetabolic and inflammatory state. Persistent hypermetabolism leads to vast catabolism and, ultimately, to multiple organ failure and death [9,22]. 

These responses present in burns also occur in other types of trauma, but they differ in their long-lasting presence in burn injury patients. The pro-inflammatory mediators (cytokines, chemokines, and acute phase proteins such as IL-1, IL-6, TNF, and IGF-1) are present in the hypermetabolic state, and they play a role in patients with burn injury (moderate and severe). 

Hypovolemic hemorrhagic shock present in trauma is secondary to blood loss and not to the leakage of fluid in the interstitial space, as in distributive shock in burns, and occurs when there is a decreased intravascular volume to the point of cardiovascular compromise [28]. In this type of shock, at the cellular level, oxygen delivery is insufficient to meet the oxygen demand of the organism, with a shift from aerobic to anaerobic metabolism due to blood loss and consequent anemia. When the ATP provision sags, the homeostasis of cells falls and the cellular membrane breaks down, apoptosis and necroptosis occur with necrosis and cellular death [28], and lactic acid, inorganic phosphates, and oxygen radicals accumulate owing to increased oxygen debt [28]. Regarding endotheliopathy, it is present both in hemorrhagic shock following trauma and in burns, producing a systemic shedding of the protective glycocalyx barriers induced by the increased oxygen debt and catecholamine surge. The persistence of hemorrhagic shock and the release of DAMPs, including mitochondrial DNA [18] and formyl peptides, produces a systemic inflammatory response. Endotheliopathy is also present in hemorrhagic shock, as in burns, with systemic shedding of the protective glycocalyx barriers induced by the increased oxygen debt and catecholamine surge. Hypovolemia and vasoconstriction cause hypoperfusion and end-organ damage. Kidney, liver, intestine, and skeletal muscles are involved, and their functions can be so impaired as to result in multiple organ failure (MOF) [28]. In the case of an excessive resuscitation with fluids, hyperdilution is present as is coagulopathy. Coagulopathy can be due to excessive resuscitation with a dilution of the oxygen-carrying capacity, increased clotting factor concentration, heat loss, and progressive worsening of acidosis [28]. There is evidence in the literature that platelet number and/or reduced platelet activity can also contribute to coagulopathy and increased mortality [28]. 

### 3.1. DAMPs and Burns

Burn injury also has an effect on the immune system [9,29,30,31]. Activated monocytes, macrophages, neutrophils, and other immune cells are able to identify endogenous factors, DAMPs [9,31], that are generated from the damaged tissue. Secondary to this thermal injury, an upregulation of inflammatory cells and innate immune markers is present in various organs such as the heart, lungs, liver, intestines, and spleen [9,32,33]. The damage associated with the DAMPs present in the injured tissue activates the inflammatory cells through toll-like receptors (TLRs) and NOD-like receptors (NLRs) [3]. The specific ligation of TLRs and NLRs activates the inflammatory pathways through the transcription factors NF-κB and multiple inflammatory mediators, such as IL-1, IL-6, IL-8, IL-18, and TNF (for a detailed description, see Auger et al., 2017 [30]). Consequently, an activation of the cycle of inflammation responsible for the systemic inflammatory response syndrome (SIRS) is present [9,13,34].

In SIRS the uncontrolled cytokine release leads to leukocyte recruitment, onset of fever or hypothermia, tachycardia, and tachypnoea [35]. Some immune functions are also compromised, such as macrophage antigen presentation or neutrophil killing of invading pathogens as well as T cell proliferation [21,36,37]. All these events produce an altered adaptive immune response and enhanced susceptibility to infections in patients at high risk for developing infectious complications due to their severe burn injuries. The microbiotas associated with the skin, as well as those associated with the respiratory tract and the gut, can be additional sources for infection in these patients. The skin barrier disruption leads to an impairment in the host defense and to an increased susceptibility to infections caused by bacteria, yeasts, fungi, and viruses. The consequent septic status can affect all parts of the body [21]. Cytokines initiate capillary leakage in most of the capillary beds. Endothelial cells play a role in this leakage and in the upregulation of the inflammatory response [21,38,39]. In addition, there is a shift to a procoagulation status that, with the capillary damage, leads to platelet adhesion and consumption of coagulation factors [40]. In fact, a reduction in the number of platelets is one of the earliest signs of sepsis [21,40,41]. The consumption of clotting factors and platelets produces disseminated intravascular coagulopathy (DIC) with a mixture of perfused and hypoperfused areas. In response to hypoxia, cells produce nitric oxide (NO) with a consequent decrease in systemic vascular resistance. Lactic acid production in hypoxic areas is another early sign of sepsis. The extensive capillary leakage associated with poor perfusion leads to relative intravascular hypovolemia and hypotension that are identifying signs of septic shock [35].

The gut is a major source of bacteria and bacterial products and plays a determinant role in sepsis pathogenesis after burn injury [42]. After a severe burn injury, immunity is reduced, the gut is hypoperfused, and this leads to impaired motility that facilitates the growth of intestinal bacteria [43]. Additionally, besides leading to tissue inflammation, hypoperfusion also produces damage by increasing intestinal permeability [43,44]. In this way, gut bacteria are allowed to pass through intestinal walls, reaching mesenteric lymph nodes, liver, and lungs and leading to “bacterial translocation”, a process that becomes recurrent when additional triggering events are added to the initial damage. Evidence is present in the literature that intestinal bacteria products, i.e., endotoxins, can cross the intestinal epithelial barrier into systemic or lymphatic circulation (for review of the intestinal mucosa modification in sepsis, see Haussner et al., 2019 [45]).

### 3.2. DAMPS and Trauma

DAMPs in trauma are released from cells that sustain damage, stimulating a sterile immune or inflammatory response [12,13,19]. In the case of dysregulation, the inflammatory and tissue repairing processes may provoke the occurrence of pathologies such as sepsis, cardiovascular disease, and neurodegenerative disease. However, in the last two decades, the concept of biological host response to trauma evolved from cytokine release, activation, and recruitment of effector cells including antigen-presenting cells to the action and role of host and alarmins [46,47,48]. In their 2018 review Relja et al. [12] described the inflammatory systemic response and the alarmins involved. In 2020, an update of the review was published, with a description of triggers, endogenous nuclear or cytosolic, that are also present in burns [13,49]. Numerous injuries can provoke DAMPs release; in the pathology of severe trauma, the generation and emission of DAMPs are not only at the origin of the clinical expression of SIRS, but also play a role in the activation of coagulation pathways. Enhanced thrombin formation increases fibrinogen cleavage and produces a stable form of fibrin clot hypercoagulability and consequent fibrinogen depletion [50]. Disseminated intravascular coagulation (DIC) is also present [51]. This disorder has a complex pathophysiology, but in any case, DAMPs play an important role in its pathogenesis [51]. In general, extracellular DNA (cfDNA), which can be released in plasma by hematopoietic cells, neutrophils, macrophages, eosinophils, tumor cells, and certain strains of bacteria, also plays a significant role [13,52]. In trauma, elevated levels of cfDNA [53,54] have both pro- and antifibrinolytic effects and can also be used to predict trauma patient’s outcome in the ICU [53,54]. Activation of neutrophils with microbial or inflammatory stimuli results in the release of neutrophil extracellular traps (NETs) [55,56]. Upon cell death or specific activation of hematopoietic and parenchymal cells, extracellular cfDNA and DNA-binding proteins (e.g., histones and HMGB1) are released into circulation [57]. DNA-binding proteins are also strongly procoagulant and are involved in the pathogenesis of DIC [13,57,58]. Another factor that can be directly influenced by DAMPs and contribute to DIC is the cellular migratory behavior through barrier loss. DAMP-induced endothelial expression of adhesion molecules facilitates leukocyte adhesion and promotes extravasation of leukocytes from the vessels into damaged tissue [59,60]. After injury or shock, the mitochondrial (mt) DAMPs such as mitochondrial DNA (mtDNA) and peptides appear in the blood [13]. They activate human leukocytes and might contribute to increasing endothelial permeability during systemic inflammation [59,61,62]. The various DAMPs “motifs” from mitochondria can act on endothelial cells and/or leukocytes via multiple pathways by enhancing leukocyte adherence to endothelial cells, activating cell–cell interactions and subsequently [13,61] resulting in conditions where inflammation pathologically increases endothelial permeability [61]. The mechanisms of a concerted action between endothelial cells and leukocytes, endothelial cell damage, leukocytes extravasation, microcirculatory disturbances, and DIC frequently lead to cell loss of parenchymal cells (MOF), but blocking or neutralizing the DAMPs with specific small molecules or antibodies ameliorate the sepsis course and lead to resolution in vivo [60]. Increased systemic levels of DAMPs have been correlated with morbidity and mortality in animals as well as in trauma patients [12,13]. In addition, mitochondrial DAMPs may be important therapeutic targets in conditions where inflammation pathologically increases endothelial permeability [61].

## 4. Discussion and Conclusions

Burns and trauma share the same mechanism of inflammation due to the release of alarmins. In trauma, the systemic inflammatory response to severe injury begins immediately after trauma damages the tissues and involves complex interactions across the hemostatic, inflammatory, endocrine, and neurological systems [12,13,46]. In both burns and trauma, the systemic inflammatory response syndrome (SIRS) is due to an uncontrolled systemic innate immune response caused by the emission of large amounts of molecules called DAMPs [16] or alarmins [19] that circulate systemically and affect the body of the patient [13]. This hyperinflammatory, sepsi-like SIRS is a severe, life-threatening condition, associated with multiple organ failure (MOF) or multiple organ dysfunction syndrome (MODS) [13]. The compensatory anti-inflammatory response syndrome (CARS) is present, giving rise to post-traumatic immunosuppression [13]. However, burns and trauma differ in many aspects. First of all, the immediate stress response in thermal injury does not recover as quickly as it does in sepsis and trauma [9]. In severe burns, the inflammatory response can be extensive and become uncontrolled, causing a condition that does not promote healing but rather leads to a generalized catabolic state that delays healing [9,15,22,63]. This peculiar response to burns, besides being associated with catabolism, increases the incidence of organ failure, infections, and death [9,63]. 

A common route is shared by trauma and burns for alarmin recognition. For example, in trauma, HMGB1, a nuclear protein passively released from the damaged tissue, acts as a binder to PRRs such as TLR4 and RAGE (receptor for advanced glycation end products). Choen in 2009 [64] reported that HGMB1 was detected in plasma within 30 min of the trauma insult. In burns, DAMPs are generated by burn-mediated tissue damage and recognized via TRLs and NOD-like receptors (NLRs) [9]. The downstream inflammatory pathway is activated along with multiple inflammatory mediators (Il-1, IL-6, Il-18, and TNF) leading to SIRS [9,13]. Moreover, the release of DAMPs is also involved in coagulation disorders and disseminated intravascular disease (DIC) [65]. 

Hypovolemic shock is present in both conditions. In burns, it is a distributive shock with increased capillary permeability, increased hydrostatic pressure across the microvasculature, protein and fluid movement from the intravascular space into the interstitial space, increased systemic vascular resistance, reduced cardiac output, and hypovolemia requiring fluid resuscitation [29]. In trauma, hemorrhagic shock is due to the loss of blood volume [65] caused by blood loss and the rate at which it occurs, where severe depletion of intravascular volume causes an inability to match the tissues’ demand for oxygen and mitochondria, no longer able to sustain aerobic metabolism, switch to anaerobic metabolism. There is an increase in heart rate and contractility to compensate for a decline of diastolic ventricular filling, which cause a decrease in cardiac output and a drop in systolic blood pressure. Other additional events that may occur are trauma-induced coagulopathy and a multifactorial biochemical response to tissue injury, with increased risk of MOF/MODS and mortality [25,26,66,67,68,69,70].

Depressed cardiac function, present in burns, develops within a few hours after the event (ebb phase), and produces a hypovolemia that is worsened by the decreased low blood flow resulting from vasoconstriction. In general, the purpose of the immune innate response that follows a major trauma is to clear the damaged tissue and restore the pre-injury status. DAMPs and PAMPs are both present and can be sensed by the so-called “first line of defense”, an inflammatory fluid-phase pathway containing proteins and/or lipids.

The balance between the pro-inflammatory and anti-inflammatory statuses mediated by cytokine release, IL-6, IL-8, IL1-Ra, and IL-10, reactive oxygen species (ROS), phagocytosis, neutrophil extracellular traps (NETs), and the killing of bacteria, has to be considered with other conditions such as the presence of endotheliopathy, electrophysiological membranes’ dysfunction, and increase of immune, coagulatory, and ROS responses. All of these conditions can lead to a further passage of PAMPs and DAMPs through the endothelium, amplifying a vicious circle between tissue injury and harmful immunological processes. In burns, the consequences related to the extent of the inflammatory response [64] must also be considered. The inflammation severity is strictly connected to the severity of the burns, i.e., their extension and depth [9]. The capillary leakage is due to the oxidative reaction, producing an increase in the levels of nitric oxide and inflammatory mediators and damaging the vascular endothelium [9,23,24].

Depressed cardiac function is also present in burns and develops within a few hours of the event (ebb phase) and it is due to oxidative stress, release of IL-6, tumor necrosis factor (TNF), and apoptosis and necrosis of the cellular components [9,27,32]. The hypovolemia caused by the depressed cardiac function is worsened by the decreased low blood flow resulting from vasoconstriction. 

In severe injury damage, DAMPs and pathogen-associated molecular patterns (PAMPs) are present [18,46,71,72]. Other molecules such as those in the serine protease system, coagulation and complement cascades [46,73], can detect DAMPs and PAMPs.

The knowledge of these mechanisms favors long-term treatment in intensive care units, preventing the development of late MODS.

Many factors may disturb the innate immune response: nosocomial infections, immunocompromising comorbidities, and microbiome perturbations. In addition, neutrophils exhibit a prolonged lifetime immediately after injury, with an increase of their autoaggressive potential [74]. On the other hand, failure to normalize posttraumatic lymphopenia is associated with a poor outcome, without any correlation with the dynamics of the leukocytosis [46,75]. 

Another important consequence of hypoperfusion, besides leading to tissue inflammation, is an increase in intestinal permeability. The damage enables the gut bacteria to pass through intestinal walls, reach mesenteric lymph nodes, liver, and lungs, and give rise to “bacterial translocation”, a process that becomes recurrent when additional triggers add to the initial damage.

The presence of hypoxia and ischemia of the intestine and of its most delicate part, the intestinal villi [76,77], in hemorrhagic shock is due to the stress response activation with a secondary splanchnic vasoconstriction. In response, the intestinal epithelial cells release DAMPs and cytokines produce an activation of leukocytes. Complement activation can also contribute to leukocyte recruitment. The IECs can also undergo a reduction of tight junctions (TJ), modification of intracellular pH, cellular swelling, apoptosis, and necrosis, ultimately resulting in remote organ injury [78]. These changes are often the cause of a lift-off phenomenon with the release of epithelial layers of intestinal villi together with the formation of edema and destruction of the regeneration base. In trauma and/or hemorrhagic shock, the damage due to oxidative stress induces a reduction of the mucosal barrier and autodigestion of mucus and IECs by intraluminal pancreatic proteases [79,80]. Another important modification is the presence of apoptosis of Paneth cells, with a decrease in the number of host-defense peptides [77,81]. All these conditions result in a “dysbiosis” that often is the result of a commensal–pathogen shift toward the gut microbiome. 

Although the paradigm explaining how gut bacteria and PAMPs enter the host’s circulation is still an unresolved issue, the trauma-induced gut driven changes in lymphatic activity can induce remote organ injuries on an immunological basis, such as cell apoptosis in the spleen, thymus, or lung (for a review of the literature, see Huber-Lang et al., 2018) [46]. 

Clinically, the attempts to develop care strategies to prevent additional tissue damage secondary to surgery have been focused on inhibiting the further generation of DAMPs and PAMPs that can trigger an escalation of a noxious immune response [46]. However, to date, knowledge of the principles of the early immune response in surgical and intensive-care strategies is incomplete.

The pathophysiological conditions that deserve further examination are numerous. An example is the different behavior of the lungs, which in burns can be affected by direct harmful conditions such as inhalation of fumes and burning particles but also suffer from systemic involvement due to innate immune system activation. Burns and hemorrhagic shock trauma can share the latter lung damage but the direct injury related to fumes inhalation is peculiar only to burn injury [46]. 

Huber–Lang et al. in 2018 [46] concluded their review by describing the possible integration of major cross-talking aspects into a clinical context. In fact, the possibility of developing control of the hyperactivation of the innate immune system could improve the outcomes in trauma patients. Therefore, in the future, it will be necessary to apply real-time monitoring of innate immune and organ responses at the “bedside”. Since innate and adaptive immune responses can differ depending on age, comorbidities, and other preexisting conditions, every effort should be made to understand and delineate post-traumatic mechanisms and the mechanism that guide them. Besides, in this era of “precision medicine” and with the current use of bioinformatics tools, these improvements may be applicable to the phenotyping of injury patterns, precision diagnosing, and treatment [46]. Finding a biomarker able to predict complications by diagnosing excessive immune system activation and inflammation is one of the present research targets [82]. The use of DAMPs as biomarkers could enable clinicians to monitor patients by measuring plasma levels [17,46,82,83]. However, to reach this target, implementation of point-of-care testing research is essential (POCT) [82]. A number of experiments with animal models on the possible use of DAMPs and PRRs were summarized in the review of Comish in 2020 [3]. Although we are still far from finding a cure to modulate and limit the excessive response to a trauma injury that may be applicable in clinical use, this may well represent the very first step on the way to new DAMP-targeted therapy approaches to modulate and stop inflammatory cascades.

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
