# Peer review of "The Role of DAMPS in Burns and Hemorrhagic Shock Immune Response: Pathophysiology and Clinical Issues. Review"

_ijms, 2021, doi:10.3390/ijms22137020_

Round 1

Reviewer 1 Report

This review is of potential interest, however, the authors should better determine, what is their aim, what hypothesis are they testing/reviewing and how they selected the literature for this review. 

The subchapter on sepsis should be deleted.  

Author Response

We deleted the sepsis sub-chapter as required.

The method of literature selection was explained in the text

The hypothesis we are testing has been better specified and the possible objectives of future research and possible achievements have been added

The manuscript was edited to meet the suggestions of referee # 2

.

Reviewer 2 Report

Comments to the Authors

Overall:

The authors have collected an important amount of references regarding burns, hemorrhage and consequent inflammatory reactions. The abstract is well designed and is directing the reader towards a comparison of the hypermetabolic responses in burns and bleeding with a specific focus on DAMPs. Unfortunately, the core of the manuscript is confusing, badly organized with irrelevant information or repeated information. It appears as a plain collection of facts without any flow.

The authors should consider a shorter article targeting the implication (DAMPs release, mechanism of action, clinical consequences and possible therapy) following hypovolemic shock for both the burned and hemorrhagic patients.

The conclusion should guide the reader how to distinguish ethology of hypermetabolic response between burns and hemorrhage and how to interpret the presence of DAMPs.

Specific comments:

The title may consider hemorrhagic shock rather than trauma which may be too broad.

  1. Introduction:

The first paragraph regarding country resources is irrelevant for this review and should not start the introduction.

Rather than a pure presentation of burn and bleeding facts the introduction should explain why DAMPs are being discussed here and are important in both categories of patients.

  1. In the comparative pathophysiology of burn and hemorrhage, the explanation of the coagulation, on page 8 describing the general process of coagulation, should not be part of this review; this more text book content. More importantly, the difference between the burn and hemorrhage should be clearly highlighted.
  2. As above, a comparison of the of DAMPs and how it is expressed in the 2 shocks should be better organized.
  3. Sepsis, although a major problem for burns should not be treated separately.
  4. Discussion and conclusion should highlight the main points of this review and not repeat more facts.

A copy with minor edits is attached to this review.

Author Response

The manuscript was edited to accommodate the required changes. The text is shorter than the first draft

We focused the manuscript on DAMPs release and mechanisms of action. As required, a section for comparing burn shock and hemorrhagic shock was introduced

The title has been changed as required

The paragraph on country resources has been omitted

We have given a different slant to the manuscript, more on the explanation than on the collection of data

Abnormalities in coagulation were omitted as required

The expression of DAMP was better emphasized

SIRS and SEPSIS have been added in the burn section

DISCUSSION and CONCLUSION have been reworked to meet the referee's requests. Recent research findings on the use of DAMPs in clinical applications, their relationships with diagnosis prognosis, and possible therapeutic use have also been added.

Round 2

Reviewer 1 Report

The authors completely rewritten their paper. I do not have any further queries. 

Author Response

Thank you for your revision

Reviewer 2 Report

Comments to the Authors

The authors did provide a major revision of their paper; this offers a better structure. However, the core of the manuscript remains unclear as far as the content of each paragraphs is concerned, and it reads still like an accumulation of facts rather than telling a story.

The authors should consider the following 1)The method should be introduced first. 2) A brief description of the patient population covered by the papers should be provided; i.e. add demographic data,  What is the severity of the burn and Hemorrhage,  are they single injury or polyinjury, is there a difference in the population? Etc.. 3) The information in paragraph #4 introduces the general etiology of burn and hemorrhage This should come before  4) Combining the introduction and the paragraph #3 to introduce DAMPS.

Then the implication of DAMPs for each injury.

Then the conclusion should a short (1/2 page) paragraph highlighting the major role of DAMPS for both injury and the strategy for treatment.

Specific comments:

The term “Trauma” is now confusing because it seems that it includes any trauma without focusing on  hemorrhage specifically.

There are numerous repeat of sentences throughout the manuscript.

Since DAMPs is a pattern, would the authors suggest an algorithm for the sequence of events in both injuries?

Some wording “loss speed” in the abstract needs clarification.

Author Response

1) We changed the title: " Similarities and Differences between Burn and  Trauma response. Overview on DAMPs, distributive shock, hemorrhagic shock, pathophysiology and clinical issues. Review."  Burns is part of Trauma. Then the two conditions are compared because they belong to the same group. Hemorrhagic shock" is the main cause of death in trauma patients and fits perfectly the aim of the review. In this way, we think the title fits the purpose of the review better.

2)The Method section is introduced first as required (Before INTRODUCTION?)

3) We add information on the population, multiple and single trauma, disease severity, and affected population

4) We have changed the position of the paragraphs as required in the paper

5) We modified the Conclusion ad Discussion Section. We have also tried to modify our work more "by telling a story" but it is not so easy to report evidence without a detailed exposition of the facts

6) As reported above the review is on Trauma (and the Innate Immune Response) of which Burns is a section. Hemorrhage is the leading cause of death in trauma and this is clearly specified in the review. It is not possible to completely overturn the purpose and objective of the review. Distributive shock, peculiar to burns, and traumatic hemorrhagic shock, they match well there is no scientific basis to eliminate the term trauma. The release of DAMPs is a consequence of a trauma not of a hemorrhage.

7) It is too early for an algorithm, many studies are needed on the topic of DAMP in trauma

8) we modified the term loss speed with bleeding speed or blood speed

Round 3

Reviewer 2 Report

Thank you for revising the manuscript.

The following are suggested:

  • Suggestion for the title “ Role of DAMPS in Burns and Hemorrhagic shock immune response. Pathophysiology and clinical issues.

  • Please review carefully the numbering of sections and correct grammatical error and unfinished sentences.

  • The term loss speed is still present in the abstract. Please correct.

Author Response

Dear Reviewer, 

Thank you for your suggestions.

We modified the title

We carefully checked spelling, grammar, and unfinished sentences

We have changed the sentence that contains "loss of speed" in the abstract for a better explanation

We checked the numbering of section

Sincerely

Desire' Pantalone MD, FACS
